# Effectiveness and Safety of Immunosuppressants and Biological Therapy for Chronic Spontaneous Urticaria: A Network Meta-Analysis

**DOI:** 10.3390/biomedicines10092152

**Published:** 2022-09-01

**Authors:** Wen-Kuang Lin, Shwu-Jiuan Lin, Woan-Ruoh Lee, Chia-Chieh Lin, Weei-Chin Lin, Hua-Ching Chang, Chi-Tsun Cheng, Jason C. Hsu

**Affiliations:** 1School of Pharmacy, Taipei Medical University, Taipei 110301, Taiwan; 2Department of Dermatology, Shuang-Ho Hospital, Taipei Medical University, New Taipei City 235041, Taiwan; 3Graduate Institute of Medical Science, School of Medicine, Taipei Medical University, Taipei 11031, Taiwan; 4Department of Pharmacy, Taipei Veterans General Hospital, Taipei 112201, Taiwan; 5Section of Hematology/Oncology, Department of Medicine and Department of Molecular and Cellular Biology, Baylor College of Medicine, Houston, TX 77030, USA; 6Department of Dermatology, Taipei Medical University Hospital, Taipei 110301, Taiwan; 7Department of Dermatology, School of Medicine, College of Medicine, Taipei Medical University, Taipei 110301, Taiwan; 8Research Center of Health Care Industry Data Science, College of Management, Taipei Medical University, Taipei 110301, Taiwan; 9International PhD Program in Biotech and Healthcare Management, College of Management, Taipei Medical University, Taipei 110301, Taiwan; 10Clinical Data Center, Office of Data Science, Taipei Medical University, Taipei 110301, Taiwan; 11Clinical Big Data Research Center, Taipei Medical University Hospital, Taipei Medical University, Taipei 110301, Taiwan

**Keywords:** immunosuppressant, biological therapy, chronic spontaneous urticaria, network meta-analysis, systematic review

## Abstract

Chronic spontaneous urticaria (CSU) is the most common phenotype of chronic urticaria. We compared treatment effects and safety profiles of the medications in patients with CSU. We searched PubMed, MEDLINE, and Web of Science for randomized control trials (RCTs), from 1 January 2000 to 31 July 2021, which evaluated omalizumab and immunosuppressants. Network meta-analyses (NMAs) were performed with a frequentist approach. Outcome assessments considered the efficacy (Dermatology Life Quality Index (DLQI) and weekly urticaria activity score (UAS7)) and tolerability profiles with evaluations of study quality, inconsistencies, and heterogeneity. We identified 14 studies which we included in our direct and indirect quantitative analyses. Omalizumab demonstrated better efficacy in DLQI and UAS7 outcomes compared to a placebo, and UAS7 assessments also demonstrated better outcomes compared to cyclosporine. Alongside this, omalizumab demonstrated relatively lower incidences of safety concerns compared to the other immunosuppressants. Cyclosporin was also associated with higher odds of adverse events than other treatment options. Our findings indicate that omalizumab resulted in greater improvements in terms of the DLQI and UAS7 with good tolerability in CSU patients compared to the other immunosuppressants.

## What is known on this topic:

EAACI/GA2LEN⁄EDF⁄WAO guidelines recommend omalizumab as a third-line add-on therapy for patients with uncontrolled CSU despite being under antihistamine treatment.

A previous NMA study illustrated the efficacy of changes in urticaria symptoms and safety outcomes from related treatments.

Evidence from head-to-head comparisons of evaluations of QoL improvement for most commonly prescribed advanced treatments is lacking.

## What this study adds:

There was an improving trend in the omalizumab group in QoL compared to methotrexate and hydroxychloroquine.

This NMA of efficacy and safety included all eligible RCTs which evaluated cyclosporin, methotrexate, hydroxychloroquine, montelukast, or omalizumab for CSU.

There were no significant differences among all treatment comparisons in terms of adverse events. However, cyclosporin and montelukast were associated with higher odds of adverse events compared to the other treatment options.

## 1. Introduction

Chronic spontaneous urticaria (CSU), also known as chronic idiopathic urticaria (CIU), is characterized by the appearance of hives, pruritus, and/or angioedema, with no obvious specific external triggers, for a duration of longer than 6 weeks [1]. CSU is the most common phenotype of chronic urticaria and affects up to 1% of the population at any given time, accounting for approximately two-thirds of cases of chronic urticaria [2]. All age groups can be affected, but one report suggested that the peak incidence is between 20 and 40 years of age, and two-fold more females suffer from this condition than males [2]. In addition, many quality of life (QoL) aspects were found to be reduced in patients with CSU, including activities of daily living, social functions, etc. [3]. Furthermore, previous publications suggested that patients with chronic urticaria had a higher risk of psychiatric disorders which ranged from 35~60% [4,5,6,7,8]. In terms of CSU management, the latest treatment guideline, developed by the European Academy of Allergy and Clinical Immunology, the Global Allergy and Asthma European Network, the European Dermatology Forum, and the World Allergy Organization (EAACI⁄GA2LEN⁄EDF⁄WAO), recommended non-sedative antihistamine medications with standard dose as first-line treatment; physicians may titrate the dose, up to four times the normal dose, as second-line treatment to manage symptomatic CSU patients despite being under first-line therapy [1]. According to the literature, however, the proportion of CSU patients with inadequate control, who have received 3~4-fold recommended doses, may be up to 50% [9,10], and these patients may receive advanced treatment, including omalizumab, cyclosporin, or other immunosuppressant agents.

Recently, Nochiawong et al. [11] conducted a network meta-analysis (NMA) to demonstrate the effectiveness of symptom changes and safety data of related medications for symptomatic CSU management, including biological therapies, immunosuppressants, and anti-inflammatory agents. Those authors collected publications with related CSU symptom evaluation scores, including the weekly urticaria activity score (UAS7) and other measurement tools. They also collected all publications which used a placebo or directly compared active comparators, such as cyclosporin and azathioprine [11]. Thus, in this study, we performed another comprehensive systematic review and NMA of randomized control trials (RCTs) with only placebo comparators to evaluate the effectiveness, which focused on UAS7, safety outcomes, and also an evaluation of improvements in quality of life (QoL), which was based on the Dermatology Life Quality Index (DLQI). Our aim was to summarize and compare the available evidence of symptomatic CSU management despite being under H1 antihistamine therapy to allow physicians to better interpret the relative efficacies and safety profiles of different treatment regimens. In addition, these data may offer policymakers additional evidence to support or revise treatment recommendations.

## 2. Methods

### 2.1. Protocol and Registration

The study was conducted and reported in line with the Preferred Reporting Items for Systematic Reviews and Meta-analyses (PRISMA)-NMA guidelines and was registered with PROSPERO (CRD42021282924).

### 2.2. Search Strategy and Study Selection

We included RCTs from electronic databases, including PubMed, MEDLINE, and Web of Science, from 1 January 2000 to 31 July 2021. In addition, we hand-searched references from included papers and relevant systematic reviews for additional relevant trials to identify potential publications.

The characteristics of included studies should meet following criteria: (1) the study design should be RCTs of CSU/CIU treatments, (2) recruited patients over the age of 12; and (3) published in the English language. Multiple-arm studies with a placebo plus H1 antihistamine agents (standard or up to 4-fold doses) and add-on active comparators with different dosages were acceptable. Two investigators independently screened all titles, abstracts, and full papers, using the eligibility criteria below, with any disagreements resolved through discussion. Two reviewers critically appraised the methodologic quality of each included study using the Jadad scale and revised Cochrane tool [12,13]. According to the revised Cochrane tool, the overall risk of bias was then classified as low, some concern, or major concern. Furthermore, based on the Jadad scale, a study with a score of <3 represented a low-quality and high-bias risk study, and a study with a score exceeding 3 was considered a high-quality trial (Appendix A). Overall assessments of imprecision, incoherence, and heterogeneity are summarized in Appendix A.

### 2.3. Eligibility Criteria

All included studies had to meet the following criteria: (1) symptomatic CSU patients despite being under H1 antihistamine treatment (standard or up to a 4-fold dose) and (2) older than 12 years of age. However, we excluded trials if (1) they were a non-human study, (2) age of recruited participants was less than 12 years old, and (3) the diagnosis was not CSU or CIU.

### 2.4. Intervention and Comparators

This study compared selected pharmacological management, including immunosuppressants such as methotrexate, cyclosporin, hydroxychloroquine, azathioprine, biological therapy, and omalizumab, and the leukotriene receptor antagonist, montelukast, in symptomatic CSU patients. We excluded studies with non-pharmacological treatment or studies that used other medications which did not fit our inclusion criteria. In terms of comparators, a placebo plus an antihistamine (a standard dose or 4-fold dose) or active comparators plus an antihistamine (a standard dose or 4-fold dose) could be included (Appendix A).

### 2.5. Study Outcomes

The primary outcome was the mean change in DLQI scores from the baseline. Secondary outcomes included the mean change in the UAS7 from the baseline and the proportion of patients experiencing at least one adverse effect.

### 2.6. Statistical Approaches

This NMA was performed using the CINeMA (Confidence in Network Meta-Analysis), and we also evaluated the certainty of the final body of evidence [14]. We employed a random-effects model framework for both continuous and binary outcomes. To estimate continuous variables, results were demonstrated by the mean difference (MD) with the 95% confidence interval (CI); in terms of binary outcomes, the effects were depicted by the odds ratio (OR) with the 95% CI.

To produce an integrated dataset, we applied a statistical method to unify the data presentation for continuous variables as the mean ± standard deviation (SD). There were two scenarios: (1) interconversion between the mean with the 95% CI and mean ± SD and (2) interconversion between the median with the range and the mean ± SD [15,16].

## 3. Results

We identified 1180 references. After removing duplicates, 854 were screened by two investigators for eligibility. In total, 14 records met the inclusion criteria and were included in the NMA (Figure 1) [17,18,19,20,21,22,23,24,25,26,27,28,29,30]. Detailed information on all studies included in the NMA is presented in Table 1, and Appendix A shows information of studies excluded from the NMA [31,32,33,34,35,36,37,38,39,40,41,42,43].

In total, 577 participants with a placebo plus H1 antihistamine treatment and 1209 participants with active comparators plus an H1 antihistamine were identified, with mean ages ranging from 32.5~46.4 years. Treatments that were eligible for the evidence network were cyclosporin, methotrexate, hydroxychloroquine, omalizumab, and montelukast. We excluded the trials with azathioprine due to insufficient outcome evaluations and no placebo comparator [31,32]. Based on the overall risk-of-bias assessment, 12 trials had a low risk of bias, 2 trials had some concerns, and no trials had a high risk of bias (Appendix A). Detailed network plots of direct evidence are also shown in Appendix A.

### 3.1. DLQI Score Improvement

Table 2 sets out direct and pooled MDs and 95% CIs for comparators. Direct comparisons with a placebo yielded a trend of QoL improvement for all active comparators. Omalizumab groups, however, with dosage of 150 mg/Q4W and 300 mg/Q4W, demonstrated significant reductions in DLQI scores of −1.900 (95% CI, −1.920 to −1.881) and −3.100 (95% CI, −3.120 to −3.081), respectively.

When comparing mixed treatment groups, omalizumab treatment showed a dose-dependent effect. The higher the dosage of omalizumab received, the more QoL improvement reported by patients. This result was also demonstrated in certain RCTs [24,26]. The effect of omalizumab at 300 mg/Q4W was greater than with methotrexate and hydroxychloroquine, but there was no significant difference.

### 3.2. UAS7 Score Improvement

Mean changes in urticarial symptoms, including hives and pruritus, are presented in Table 3. The largest change was observed in those using omalizumab treatment with an MD of −12.0 (95% CI, −19.988 to −4.012) compared to a placebo. In addition, the cyclosporin group also showed a great reduction in the UAS7 with an MD of −10.4 (95% CI, −18.587 to −2.213). There was no significant difference in the montelukast group (MD, −0.366; 95% CI, −5.315 to 4.583), but it still demonstrated an improving trend. There was an interesting finding that the efficacy of the methotrexate group was weaker than a placebo. This result was also reported by a previous NMA publication [11].

In addition, there were similar results between the DLQI and UAS7, suggesting that omalizumab has a dose-dependent effect. Our results are consistent with reports of real-world evidence [44,45]. There was no significant difference between omalizumab at 300 mg/Q4W and cyclosporin, but it still indicated an improving trend in the group of omalizumab at 300 mg/Q4W. Network comparisons, on the other hand, suggested a significant improvement with omalizumab at 300 mg/Q4W compared to montelukast and methotrexate.

### 3.3. Adverse Events

We found no significant difference among all treatment comparisons in terms of at least one adverse event. Nevertheless, we noticed some interesting trends. Cyclosporin and montelukast, for instance, were associated with higher odds of adverse events than other treatment options. The NMA league table of adverse events is shown in Table 4.

## 4. Discussion

There are now several approved medications available for CSU, including H1 antihistamines and omalizumab. In clinical settings, physicians may use other immunosuppressants, including cyclosporin, methotrexate, azathioprine, and hydroxychloroquine. In the absence of head-to-head studies, we conducted this NMA to evaluate the comparative efficacy and tolerability of biotherapeutic agents and immunosuppressants in symptomatic CSU patients in addition to standard treatment.

Compared to immunosuppressants, omalizumab was generally associated with greater reductions in DLQI and UAS7 scores, while also demonstrating a low incidence of adverse effects. Cyclosporin was recommended as fourth-line treatment for uncontrolled CSU patients by EAACI⁄GA2LEN⁄EDF⁄WAO guidelines. In clinical settings, physicians may try to manage refractory CSU patients using cyclosporin or other immunosuppressants due to economic considerations. Indeed, there are few studies that reviewed the safety and efficacy of cyclosporin. Kulthanan et al. [46] conducted a systematic review and meta-analysis demonstrating that cyclosporin was an effective treatment option for CSU patients. However, adverse events occurred in more than half of patients treated with a moderate dose. Our analysis indicated similar results, that cyclosporin could lead to increased incidences of adverse effects compared to omalizumab. After reviewing the clinical evidence available to date, we conclude that cyclosporin can provide benefits to patients with CSU, despite a higher incidence of adverse effects.

Our analysis suggests that the effectiveness of omalizumab in CSU management is dose dependent. This observation is supported by an RCT, by clinical evidence, and also by some up-dosing studies in a real-world setting [44,45]. The pathophysiology of CSU is associated with immunoglobulin E (IgE), IgG, or related autoreactivity factors which activate mast cells and basophils, leading to CSU symptoms [47]. Mast cells and basophils can be stabilized through neutralizing the serum free form of IgE and downregulating the high-affinity IgE receptor. The most important pharmacological effect of omalizumab is IgE receptor downregulation, and this action occurs by neutralizing the serum-free form of IgE to reduce the level to near zero [48]. Our results and the scientific theory suggest that omalizumab confers superior efficacy and a good safety profile compared to other immunosuppressants. On the other hand, Turk et al. suggested that CSU patients who are partial responders to omalizumab may benefit from up-dosing of omalizumab or shortening the treatment interval in different conditions [49]. This may be explained by Chang et al.’s study, which demonstrated that an adequate anti-IgE dose can maintain the serum-free level of IgE to near zero, leading to subsequent pharmacological effects to achieve symptom improvement [48]. Therefore, CSU patients who respond to an increasing dose might have higher blood IgE levels or body weight than CSU patients who are responders to the approved dose of omalizumab.

Another intriguing finding was that the efficacy of methotrexate was found to be weaker than a placebo in CSU management. In fact, this result was also supported by a previous systematic review and meta-analysis, which found that there was no significant benefit of adding methotrexate to an antihistamine in refractory urticaria management [50]. Due to limited adequate references of methotrexate in CSU management, there was only one clinical trial that met our inclusion criteria for inclusion in our quantitative analysis. That was the same circumstance we encountered in the group of omalizumab 600 mg/Q4W; there was only one arm in one study which tested this specific regimen. The small sample size limits the validity of these results.

Currently, there are potential compounds under investigation and clinical trials. Ligelizumab, an anti-IgE treatment, has higher binding affinity to IgE receptors than omalizumab. The efficacy of ligelizumab in CSU management is under investigation by phase 3 clinical trials [51,52], but public announcement by a pharmaceutical company provided some observations that ligelizumab demonstrated superiority versus placebo but not versus omalizumab. Other new ingredients under development include anti-interleukin 4α receptor (dupilumab) [53], anti-interleukin 5 receptor (benralizumab) [54], anti-interleukin 5 (mepolizumab) [55], Janus kinase inhibitor (TLL018) [56], Bruton tyrosine kinase inhibitors (remibrutinib and rilzabrutinib) [57,58], humanized anti-KIT IgG1 monoclonal antibody (barzolvolimab) [59], and anti-thymic stromal lymphopoietin (Tezepelumab) [60].

## 5. Strengths and Limitations

To our knowledge, this is the first NMA study not only to analyze a questionnaire of the QoL, which is the focus of the DLQI, as the primary endpoint, but also to focus on immunosuppressants and biologics on the market. We conducted a comprehensive and robust systematic review and NMA to produce some clinical insights.

There are some limitations to our review and NMA. First, statistical calculations were implemented to maintain consistency in the data presentation when conducting the analysis. This may have over- or under-estimated the effects of related medications. Second, smaller sample sizes in individual studies might have led to a high level of heterogeneity by the large variation in the magnitude of the effect across all included studies. Finally, we only included RCT studies to conduct the NMA. As a result, our findings may not represent all refractory CSU patients due to low external validity.

## 6. Conclusions

In conclusion, our results suggest that biological treatment with omalizumab produced a greater improvement in the DLQI and UAS7 in CSU patients compared to other immunosuppressants. The safety profile of omalizumab found in this NMA was similar or superior to those of other immunosuppressants. These NMA results on treatment options in CSU can help guide our clinical practice and serve as evidence for policymakers for revising treatment recommendations.

## Figures and Tables

**Figure 1 biomedicines-10-02152-f001:**
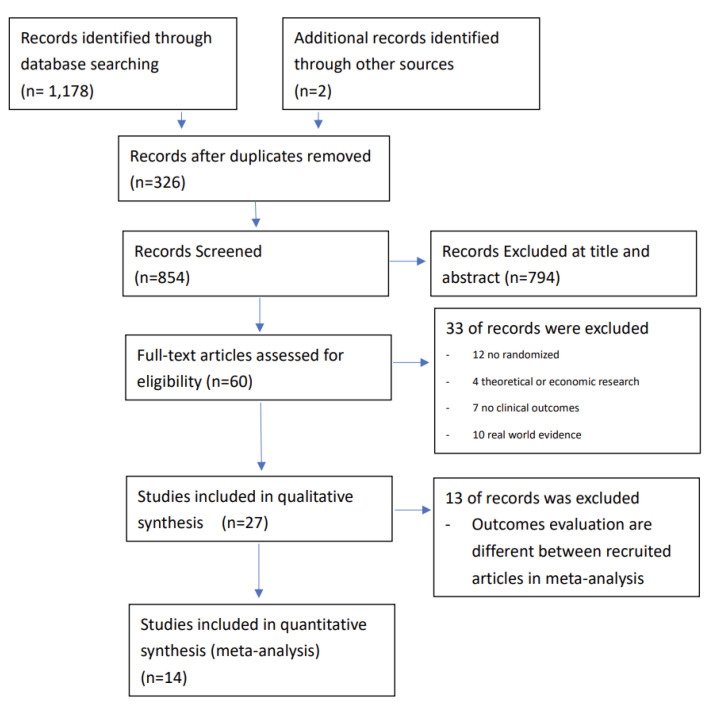
Flow diagram in the PRISMA format showing identification of the relevant literature.

**Table 1 biomedicines-10-02152-t001:** Characteristics of included studies.

Source	Study Design (No. of Patients)	Age, Mean (SD), yr	Gender	Comparator	Treatment Outcome Evaluation			Jadad Score	Overall RoB
					DLQI	UAS7	Adverse Effects		
Grattan et al. [17] 2000	RCT (*n* = 30)	Intervention: 32.5 (range: 19~72) Placebo: 33.5 (range: 23~60)	Intervention: Female: 80% Placebo: Female: 80%	CsA 4 mg/kg/day Placebo		V	V	3	Low risk
Sharma et al. [18] 2014	RCT (*n* = 29)	Intervention: 34.2 (10.4) Placebo: 30.1 (10.1)	Intervention: Female: 57% Placebo: Female: 60%	Methotrexate 15 mg/week, orally Placebo		V	V	4	Low risk
Leducq et al. [19] 2020	RCT (*n* = 75)	Intervention: 46.4 (12.6) Placebo: 45.0 (13.2)	Intervention: Female: 71.8% Placebo: Female: 80.6%	Methotrexate 0.02 mg/kg/week, orally Placebo	V			2	Low risk
Erbagci et al. [20] 2002	SB, randomized placebo-controlled crossover study (*n* = 30)	Intervention: 42.5 (range: 30~56) Placebo: 43 (range: 30~56)	Intervention: Female: 66.7% Placebo: Female: 73.3%	Montelukast 10 mg/day Placebo		V	V	2	Some concern
Sarkar et al. [21] 2017	RCT (*n* = 120)	Intervention: 32.9 (range: 18~55) Placebo: 36.2 (range: 19~71)	Intervention: Female: 64.7% Placebo: Female: 59.6%	Montelukast 10 mg/day Placebo		V		4	Low risk
Maurer et al. [22] 2011	RCT (*n* = 49)	Intervention: 39.1 (range: 24~57) Placebo: 42.3 (range: 20~69)	Intervention: Female: 70.4% Placebo: Female: 86.4%	Omalizumab Placebo		V	V	3	Low risk
Saini et al. [23] 2011	RCT (*n* = 90)	Intervention: OMA (600): 40.0 (11.1) OMA (300): 42.9 (15.7) OMA (75): 38.8 (15.5) Placebo: 41.2 (16.2)	Intervention: OMA (600): 57.1% OMA (300): 68% OMA (75): 65.3% Placebo: 81%	OMA (600) OMA (300) OMA (75) Placebo		V		4	Low risk
Maurer et al. [24] 2013	RCT (*n* = 322)	Intervention: OMA (300): 44.3 (13.7) OMA (150): 43.0 (13.2) OMA (75): 39.7 (15.0) Placebo: 43.1 (12.5)	Intervention: OMA (300): 80% OMA (150): 79% OMA (75): 74% Placebo: 70%	OMA (300) OMA (150) OMA (75) Placebo	V	V	V	3	Low risk
Kaplan et al. [25] 2013	RCT (*n* = 336)	Intervention: OMA (300): 42.7 (13.9) Placebo: 44.3 (14.7)	Intervention: OMA (300): 73.8% Placebo: 66.3%	OMA (300) Placebo	V	V	V	3	Low risk
Saini et al. [26] 2015	RCT (*n* = 318)	Intervention: 41.4 (14.1) Placebo: 40.4 (15.6)	Intervention: OMA (300): 74.1% OMA (150): 80% OMA (75): 71.4% Placebo: 65%	OMA (300) OMA (150) OMA (75) Placebo	V	V	V	4	Low risk
Staubach et al. [27] 2016	RCT (*n* = 91)	Intervention: 44.9 (range: 20~73) Placebo: 41.4 (range: 20~61)	Intervention: OMA (300): 68.2% Placebo: 70.2%	OMA (300) Placebo	V	V		3	Low risk
Hide et al. [28] 2017	RCT (*n* = 218)	Intervention: OMA (300): 44.6 (14.9) OMA (150): 43.6 (12.2) Placebo: 42.5 (14.3)	Intervention: OMA (300): 54.8% OMA (150): 60.6% Placebo: 64.9%	OMA (300) OMA (150) Placebo	V	V	V	3	Low risk
Metz et al. [29] 2017	RCT (*n* = 40)	Intervention: OMA (300): 37.5 (11) Placebo: 41.1 (8)	Intervention: OMA (300): 90% Placebo: 80%	OMA (300) Placebo		V		3	Low risk
Boonpiyathad et al. [30] 2017	SB, RCT (*n* = 55)	Intervention: 33 (12.1) Placebo: 33.9 (11.9)	Intervention: Female: 87.5% Placebo: Female: 83.3%	HCQ Placebo	V		V	2	Some concern

CsA, cyclosporin; DLQI, Dermatology Life Quality Index; HCQ, hydroxychloroquine; OMA, omalizumab; OMA (75), omalizumab 75 mg/month; OMA (150), omalizumab 150 mg/month; OMA (300), omalizumab 300 mg/month; OMA (600), omalizumab 600 mg/month; RCT, randomized controlled trial; RoB, risk of bias; SB, single-blind; SD, standard deviation; UAS7, weekly urticaria activity score. “Treatment outcome evaluation” is the main effectiveness and safety outcomes of analyzed studies.

**Table 2 biomedicines-10-02152-t002:** League table of the Dermatology Life Quality Index (DLQI) assessment.

A1H1 + OMA (150)	1.200 (1.181, 1.220)	−1.545 (−2.740, −0.351)	0.160 (−3.663, 3.983)	−0.250 (−4.280, 3.780)	−1.900 (−1.920, −1.881)
−1.200 (−1.220, −1.181)	A1H1 + OMA (300)	−2.745 (−3.940, −1.551)	−1.040 (−4.863, 2.782)	−1.450 (−5.481, 2.580)	−3.100 (−3.120, −3.081)
1.545 (0.351, 2.740)	2.745 (1.551, 3.940)	A1H1 + OMA (75)	1.705 (−2.300, 5.710)	1.295 (−2.909, 5.499)	−0.355 (−1.549, 0.840)
−0.160 (−3.983, 3.663)	1.040 (−2.782, 4.863)	−1.705 (−5.710, 2.300)	HCQ + A1H1	−0.410 (−5.965, 5.145)	−2.060 (−5.883, 1.763)
0.250 (−3.780, 4.280)	1.450 (−2.580, 5.481)	−1.295 (−5.499, 2.909)	0.410 (−5.145, 5.965)	MTX +A1H1	−1.650 (−5.680, 2.380)
1.900 (1.881, 1.920)	3.100 (3.081, 3.120)	0.355 (−0.840, 1.549)	2.060 (−1.763, 5.883)	1.650 (−2.380, 5.680)	Placebo + A1H1

Data are expressed as the mean difference (95% confidence interval) for the DLQI. Comparisons of the lower left triangle should be read from left to right, but the comparisons between treatments in the upper right triangle should be read from right to left. In the upper triangle, a value of <0 favors the treatment in the corresponding row. Light green indicates significant results. A1H1, H1-antihistamine; HCQ, hydroxychloroquine; MTX, methotrexate; OMA (75), omalizumab 75 mg/Q4W; OMA (150), omalizumab 150 mg/Q4W; OMA (300), omalizumab 300 mg/Q4W.

**Table 3 biomedicines-10-02152-t003:** League table of the weekly urticaria activity scale (UAS7) assessment.

A1H1 + OMA	−5.842 (−14.148, 2.464)	−1.160 (−9.364, 7.043)	−4.846 (−14.760, 5.069)	−8.734 (−17.229, −0.240)	−1.600 (−13.038, 9.838)	−11.634 (−21.031, −2.237)	−16.130 (−24.977, −7.283)	−12.000 (−19.988, −4.012)
5.842 (−2.464, 14.148)	A1H1 + OMA (150)	4.681 (2.390, 6.973)	0.996 (−5.121, 7.114)	−2.893 (−6.002, 0.216)	4.242 (−4.255, 12.739)	−5.792 (−11.239, −0.345)	−10.288 (−14.718, −5.858)	−6.158 (−8.434, −3.883)
1.160 (−7.043, 9.364)	−4.681 (−6.973, −2.390)	A1H1 + OMA (300)	−3.685 (−9.585, 2.215)	−7.574 (−10.493, −4.655)	−0.440 (−8.836, 7.957)	−10.473 (−15.763, −5.184)	−14.970 (−19.204, −10.735)	−10.840 (−12.706, −8.974)
4.846 (−5.069, 14.760)	−0.996 (−7.114, 5.121)	3.685 (−2.215, 9.585)	A1H1 + OMA (600)	−3.889 (−9.941, 2.163)	3.245 (−6.829, 13.320)	−6.788 (−14.468, 0.891)	−11.284 (−18.280, −4.289)	−7.154 (−13.027, −1.282)
8.734 (0.240, 17.229)	2.893 (−0.216, 6.002)	7.574 (4.655, 10.493)	3.889 (−2.163, 9.941)	A1H1 + OMA (75)	7.134 (−1.547, 15.816)	−2.899 (−8.630, 2.831)	−7.396 (−12.170, −2.621)	−3.266 (−6.155, −0.376)
1.600 (−9.838, 13.038)	−4.242 (−12.739, 4.255)	0.440 (−7.957, 8.836)	−3.245 (−13.320, 6.829)	−7.134 (−15.816, 1.547)	CsA 4mg/kg/day + A1H1	−10.034 (−19.600, −0.468)	−14.530 (−23.556, −5.504)	−10.400 (−18.587, −2.213)
11.634 (2.237, 21.031)	5.792 (0.345, 11.239)	10.473 (5.184, 15.763)	6.788 (−0.891, 14.468)	2.899 (−2.831, 8.630)	10.034 (0.468, 19.600)	Montelukast 10mg +A1H1	−4.496 (−10.737, 1.744)	−0.366 (−5.315, 4.583)
16.130 (7.283, 24.977)	10.288 (5.858, 14.718)	14.970 (10.735, 19.204)	11.284 (4.289, 18.280)	7.396 (2.621, 12.170)	14.530 (5.504, 23.556)	4.496 (−1.744, 10.737)	MTX 15mg/wk + A1H1	4.130 (0.329, 7.931)
12.000 (4.012, 19.988)	6.158 (3.883, 8.434)	10.840 (8.974, 12.706)	7.154 (1.282, 13.027)	3.266 (0.376, 6.155)	10.400 (2.213, 18.587)	0.366 (−4.583, 5.315)	−4.130 (−7.931, −0.329)	Placebo + A1H1

Data are expressed as the mean difference (95% confidence interval) for the UAS7. Comparisons of the lower left triangle should be read from left to right, but comparisons between treatments in the upper right triangle should be read from right to left. In the upper triangle, a value of <0 favors the treatment in the corresponding row. Light green indicates significant results. A1H1, H1-antihistamine; CsA, cyclosporin; HCQ, hydroxychloroquine; MTX, methotrexate; OMA, omalizumab; OMA (75), omalizumab 75 mg/Q4W; OMA (150), omalizumab 150 mg/Q4W; OMA (300), omalizumab 300 mg/Q4W; OMA (600), omalizumab 600 mg/Q4W.

**Table 4 biomedicines-10-02152-t004:** League table of adverse event assessments.

A1H1 + OMA	0.900 (0.000, 2638.590)	0.844 (0.000, 1438.132)	0.907 (0.000, 5584.220)	0.803 (0.001, 747.400)	0.849 (0.000, 20381.848)	0.574 (0.001, 602.809)	0.602 (0.001, 664.078)	0.778 (0.002, 276.083)
1.111 (0.000, 3259.405)	A1H1 + OMA (150)	0.938 (0.007, 117.731)	1.008 (0.002, 451.240)	0.892 (0.001, 560.315)	0.943 (0.000, 17422.191)	0.638 (0.001, 455.229)	0.669 (0.001, 502.955)	0.864 (0.004, 193.040)
1.185 (0.001, 2020.095)	1.066 (0.008, 133.874)	A1H1 + OMA (300)	1.075 (0.003, 341.108)	0.952 (0.003, 300.877)	1.006 (0.000, 12018.471)	0.680 (0.002, 248.092)	0.713 (0.002, 275.724)	0.922 (0.010, 88.979)
1.103 (0.000, 6789.958)	0.992 (0.002, 444.122)	0.930 (0.003, 295.243)	A1H1 + OMA (75)	0.885 (0.001, 1366.216)	0.936 (0.000, 31869.350)	0.633 (0.000, 1093.020)	0.664 (0.000, 1200.148)	0.857 (0.001, 544.517)
1.245 (0.001, 1159.449)	1.121 (0.002, 703.663)	1.051 (0.003, 332.287)	1.130 (0.001, 1743.239)	CsA 4mg/kg/day +A1H1	1.057 (0.000, 7879.345)	0.715 (0.004, 119.033)	0.750 (0.004, 133.446)	0.969 (0.029, 32.069)
1.178 (0.000, 28302.347)	1.060 (0.000, 19584.623)	0.994 (0.000, 11879.862)	1.069 (0.000, 36399.124)	0.946 (0.000, 7052.941)	HCQ + A1H1	0.676 (0.000, 5532.527)	0.709 (0.000, 6027.575)	0.916 (0.000, 3339.579)
1.742 (0.002, 1830.053)	1.568 (0.002, 1118.899)	1.470 (0.004, 536.250)	1.580 (0.001, 2729.299)	1.399 (0.008, 232.944)	1.479 (0.000, 12095.636)	Montelukast 10mg + A1H1	1.049 (0.005, 218.700)	1.355 (0.032, 56.486)
1.661 (0.002, 1832.433)	1.495 (0.002, 1123.495)	1.402 (0.004, 541.639)	1.507 (0.001, 2723.846)	1.334 (0.007, 237.365)	1.410 (0.000, 11977.678)	0.953 (0.005, 198.780)	MTX 15mg/wk + A1H1	1.292 (0.028, 58.962)
1.286 (0.004, 456.596)	1.157 (0.005, 258.475)	1.085 (0.011, 104.773)	1.166 (0.002, 740.703)	1.033 (0.031, 34.192)	1.091 (0.000, 3977.465)	0.738 (0.018, 30.772)	0.774 (0.017, 35.343)	Placebo + A1H1

Data are expressed as the odds ratio (95% confidence interval) for adverse effects. Comparisons of the lower left triangle should be read from left to right, but comparisons between treatments in the upper right triangle should be read from right to left. In the upper triangle, a value of >1 indicates that the treatment in the corresponding row tends to produce related adverse events. A1H1, H1-antihistamine; CsA, cyclosporin; HCQ, hydroxychloroquine; MTX, methotrexate; OMA, omalizumab; OMA (75), omalizumab 75 mg/Q4W; OMA (150), omalizumab 150 mg/Q4W; OMA (300), omalizumab 300 mg/Q4W.

## Data Availability

Data collected and extracted into Excel sheets used in our analyses are available upon reasonable request from the corresponding author.

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
