# Peer review of "Effectiveness and Safety of Immunosuppressants and Biological Therapy for Chronic Spontaneous Urticaria: A Network Meta-Analysis"

_biomedicines, 2022, doi:10.3390/biomedicines10092152_

Round 1

Reviewer 1 Report

The authors reported the results of a meta-analysis evaluating the effectiveness and safety of biologics and immunosuppressant in chronic spontaneous urticaria. The manuscript is interesting. However, I have few comments. - English language should be revised - Efficacy and Safety: a table comparing the main effectiveness and safety outcomes (also including main AEs) of the analyzed studies should be inserted - Strengths and Limitations: Strengths and Limitations should be discussed in a separate section - A paragraph briefly introducing promising ongoing studies on new drugs for CSU management should be inserted in order to provide readers with a comprehensive overview of future perspectives.

Author Response

Effectiveness and safety of immunosuppressants and biological therapy for chronic spontaneous urticaria: a network meta-analysis

Responses to Reviewer 1 Comments

We appreciate the thoughtful comments made by the reviewer and believe that our responses (detailed below) have addressed his/her recommendations and have substantially improved the manuscript. This entire manuscript has been reviewed by a native English speaker to improve the English language. Our revisions are highlighted using the track change function in the manuscript. Many thanks for the reviewer’s comments.

Reviewer 1

The authors reported the results of a meta-analysis evaluating the effectiveness and safety of biologics and immunosuppressant in chronic spontaneous urticaria. The manuscript is interesting. However, I have few comments.

  1. English language should be revised.

Response: We appreciate the reviewer’s suggestion. The revision has been reviewed by native English speaker to improve the English language.

  1. Efficacy and Safety: a table comparing the main effectiveness and safety outcomes (also including main AEs) of the analyzed studies should be inserted.

Response: We summarized the characteristics of included studies, including effectiveness and safety outcomes, in table 1. We also added an annotation to allow readers to easily realize the effectiveness and safety outcomes of analyzed studies as follow:

“"Treatment outcome evaluation" are the main effectiveness and safety outcomes of analyzed studies.” (in Table 1)

  1. Strengths and Limitations: Strengths and Limitations should be discussed in a separate section

Response: We appreciate the reviewer's suggestion. We have added a "Strengths and Limitations" section, and put the strengths and limitations of this study under the new section. (in the Discussion section)

  1. A paragraph briefly introducing promising ongoing studies on new drugs for CSU management should be inserted in order to provide readers with a comprehensive overview of future perspectives.

Response: We have added one paragraph to depict some potential/ new mechanism of action of therapies as follow:

“Nowadays, there are potential compounds under investigations and clinical trials. Ligelizumab, anti-IgE treatment, has higher binding affinity to IgE receptors than omalizumab. The efficacy of ligelizumab in CSU management is under investigation by phase 3 clinical trials51, 52 but public announcement by pharmaceutical company provided some observations that ligelizumab demonstrated superiority versus placebo but not versus omalizumab. Other new ingredients under development include anti-interlukin 4α receptor (dupilumab),53 anti-interlukin 5 receptor (benralizumab),54 anti-interlukin 5 (mepolizumab),55 Janus kinase inhibitor (TLL018),56 Bruton tyrosine kinase inhibitors (remibrutinib and rilzabrutinib),57, 58 humanized anti-KIT IgG1 monoclonal antibody (barzolvolimab),59 and anti-thymic stromal lymphopoietin (Tezepelumab).60” (in the Discussion section, para 5)

Reviewer 2 Report

This is a meta-analysis on the therapy for CSU.

It is interesting. I have just few comments:

Line 23: why do you say related immunosuppressants? the owrd related is not understandble for me

Line 29: compared to placebo is possibly better than to a placebo

Line 143: i have possibily not understood, but i read that you idetified 1147 references, and 828 were screened. but than in the figure 1 I read 1178 records identified, and 854 records cscreened. there is a mistake?

Line 248: what do you mean with the expression "scientific theory"?

In the title i read immunosuppressant and biological therapy......, but in the paper i have read just data about omalizumab, not othr biological therapies... didn't you find records with other biological therapies to compare? do you think it could be useful to talk about this aspect in the discussion?

Author Response

Effectiveness and safety of immunosuppressants and biological therapy for chronic spontaneous urticaria: a network meta-analysis

Responses to Reviewer 2 Comments

We appreciate the thoughtful comments made by the reviewer and believe that our responses (detailed below) have addressed his/her recommendations and have substantially improved the manuscript. This entire manuscript has been reviewed by a native English speaker to improve the English language. Our revisions are highlighted using the track change function in the manuscript. Many thanks for the reviewer’s comments.

Referee 2

This is a meta-analysis on the therapy for CSU. It is interesting. I have just few comments:

  1. Line 23: why do you say related immunosuppressants? the word related is not understandble for me.

Response: Many thanks for reviewer’s suggestion. We have mentioned “related immunosuppressants” to present that we included multiple immunosuppressants. To avoid misunderstanding and confusing, we have deleted the word, “related”. The revised sentence is as follow:

“We searched PubMed, MEDLINE, and Web of Science for randomized control trials (RCTs), from 1 Jan. 2000 to 31 July 2021, which evaluated omalizumab and immunosuppressants.” (in Abstract)

  1. Line 29: compared to placebo is possibly better than to a placebo.

Revised wording is as follow:

Response: “Omalizumab demonstrated better efficacy in DLQI and UAS7 outcomes compared with a placebo; and UAS7 assessments also demonstrated better outcomes compared with cyclosporine.” (in Abstract)

  1. Line 143: i have possibily not understood, but i read that you idetified 1147 references, and 828 were screened. but than in the figure 1 I read 1178 records identified, and 854 records cscreened. there is a mistake?

Response: Many thanks for reviewer’s reminder. After checking the original sources, that should be a typo. Revised sentence is as follow:

“We identified 1180 references. After removing duplicates, 854 were screened by two investigators for eligibility.” (in the Results section, para 1)

  1. Line 248: what do you mean with the expression "scientific theory"?

Response: The scientific theory represents the role of IgE and mast cells in CSU pathophysiology, the mechanism of action of omalizumab to confer the benefits in CSU patient management. These are including fundamental pharmacology of omalizumab (neutralize free form IgE and down-regulation of IgE receptor) and HCPs titrate the dose of omalizumab to make free IgE level near to zero in some CSU patients with high baseline IgE level. We mentioned these from line 240 to 248 as follows:

“Our analysis suggests that the effectiveness of omalizumab in CSU management is dose-dependent. This observation is supported by an RCT, by clinical evidence, and also by some up-dosing studies in a real-world setting.44,45 The pathophysiology of CSU is associated with immunoglobulin E (IgE), IgG, or related autoreactivity factors which activate mast cells and basophils, leading to CSU symptoms.47 Mast cells and basophils can be stabilized through neutralizing the serum free-form of IgE and downregulating the high-affinity IgE receptor. The most important pharmacological effect of omalizumab is IgE receptor downregulation, and this action occurs by neutralizing the serum free-form of IgE to reduce the level to near zero.48” (in the Discussion section, para 3)

  1. In the title i read immunosuppressant and biological therapy......, but in the paper i have read just data about omalizumab, not othr biological therapies... didn't you find records with other biological therapies to compare? do you think it could be useful to talk about this aspect in the discussion?

Response: Omalizumab is the only available biological therapy for CSU treatment currently. Therefore, this study only focused on omalizumab. Other biologics which are under investigations, such as ligelizumab and dupilumab, were not included in this study.

Round 2

Reviewer 2 Report

i have not more comments